# DESIGNING AND USING GOAL-CONDITIONED TOOLS

## ABSTRACT

When limited by their own morphologies, humans and some species of animals have the remarkable ability to use objects from the environment towards accomplishing otherwise impossible tasks. Embodied agents might similarly unlock a range of additional capabilities through tool use. Recent techniques for jointly optimizing morphology and control via deep learning output effective solutions for tasks such as designing locomotion agents. But while designing a single-goal morphology makes sense for locomotion, manipulation involves a wide variety of strategies depending on the task goals at hand. An agent must be capable of *rapidly prototyping* specialized tools for different goals. Therefore, we propose the idea of learning a *designer policy*, rather than a single design. A designer policy is conditioned on task goals, and outputs a design for a tool that helps solve the task. A design-agnostic controller policy can then perform manipulation using these tools. In this work, we introduce a reinforcement learning framework for learning these policies. Through simulated manipulation tasks, we show that this framework is more sample efficient than black-box optimization methods in multi-goal settings. It can also perform zero-shot interpolation or finetuning to tackle previously unseen goals. Finally, we demonstrate that our framework allows tradeoffs between the complexity of design and control policies when required by practical constraints. Additional task visualizations can be found at this link: `https://tool-design-iclr-2023.github.io/`.

## 1 INTRODUCTION

Humans and animals are able to make use of tools to solve manipulation tasks when they are constrained by their own morphologies. For example, when an item has been lost below the sofa, one might quickly deduce that a long stick will help them retrieve it. Chimpanzees have been observed using tools to access termites as food and hold water (Goodall, 1964), and cockatoos are able to create stick-like tools by cutting shapes from wood (Auersperg et al., 2016). To flexibly and resourcefully accomplish a range of tasks comparable to humans, embodied agents should also be able to leverage tools. However, while any object in a human or robot's environment is a potential tool, these objects often need to be correctly selected or combined to form a useful aid for the task goal at hand. For this reason, we investigate not only how agents can perform control using tools, but also how they can *design* appropriate tools when presented with a particular task goal, such as a target position or object location.

For an embodied agent to design and use tools in realistic environments with minimal supervision, it must be able to efficiently learn design and control policies with reward signals specified based only on task completion. Furthermore, it should form specialized tools based on the task goal at hand, as shown in Figure 1. Finally, it should be able to work with the materials it has available, rather than attempting to create potentially unrealizable designs.

Without detailed supervision, how can an agent acquire effective policies for both tool design and control? The combined space of potential designs and control policies grows exponentially even for simple tasks, and the majority of candidate tools and trajectory executions may not make any progress towards task completion. As a result, zeroth-order optimization techniques like evolutionary strategies and naive reinforcement learning approaches require many samples from the environment to find solutions. Prior works have studied joint learning of agent morphologies and control policies for locomotion tasks (Pathak et al., 2019; Luck et al., 2019; Hejna et al., 2021; Gupta et al., 2021), and methods leveraging graph neural networks (GNN)s have shown promising performance

improvements using just task rewards as supervision (Yuan et al., 2022). However, these approaches optimize designs for a generic goal, such as maintaining balance or forward speed. Designing a single-goal morphology is suited for locomotion, but manipulation requires a range of strategies depending on the given task. An agent must be capable of rapidly prototyping specialized tools for different manipulation goals.

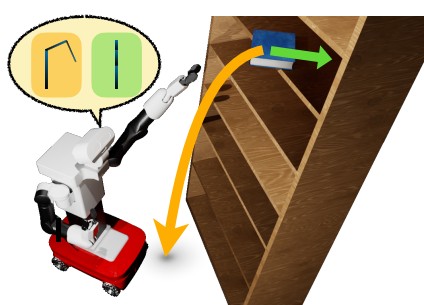

In this work, we tackle the challenges of learning design and control solely from task progress rewards by leveraging recent work in joint morphology and control optimization for locomotion agents, which performs RL using a multi-stage Markov decision process (MDP) combined with graph neural network (GNN) policies and value functions to achieve improved sample efficiency and performance in the joint learning setting. This, combined with a simple chain-link tool design parameterization, allows us to perform efficient learning of designer and controller policies together in a high-dimensional combined space. We train these policies on multiple goal settings for each task so that they can produce designs best suited to each goal and manipulate tools with varying geometries. Lastly, we investigate how tunable parameters can control the trade-offs between the complexity of design and manipulation under resource constraints.

Figure 1: An agent may need to design and use different tools to fetch a high-up book (orange) or push it into the bookshelf (green). Therefore, it should *rapidly prototype* tools for the tasks at hand.

Our main contribution is a learning framework for embodied agents to design and use rigid tools for manipulation tasks. We leverage a multi-stage reinforcement learning pipeline to learn goal-specific tools in addition to manipulation policies that can perform control with a range of tools. We demonstrate that this approach can jointly learn these policies in a sample-efficient manner in a variety of sparse reward manipulation tasks, outperforming zeroth-order stochastic optimization approaches. By introducing a tradeoff parameter between the complexity of design and control components, our approach allows us to adjust the learned components to fit resource and environmental constraints, such as available materials or energy costs. To the best of our knowledge, this work is the first that studies learning goal-dependent tool design and control without any prior knowledge about the task.

## 2 RELATED WORK

**Computational approaches to agent design.** Many works have studied the problem of optimizing the design of robotic agents and end-effectors via model-based optimization (Kawaharazuka et al., 2020; Allen et al., 2022), generative modeling (Wu et al., 2019; Ha et al., 2020), evolutionary strategies (Hejna et al., 2021), stochastic optimization (Exarchos et al., 2022), or reinforcement learning (Li et al., 2021). These methods provide feedback to the design procedure by having the agent execute predefined trajectories or perform motion planning. In contrast, we aim to *jointly* learn control policies along with designing tool structures. In settings where the desired design is known but must be assembled from subcomponents, geometry (Nair et al., 2020) and reinforcement learning (Ghasemipour et al., 2022) have been used to compose objects into tools.

**Learning robotic tool use.** Several approaches have been proposed for empowering robots to learn to use tools. Affordance learning is one common paradigm (Fang et al., 2018; Brawer et al., 2020; Xu et al., 2021). Noguchi et al. (2021) integrate tool and gripper action spaces in a Transporter-style framework. Learned or simulated dynamics models (Allen et al., 2019; Xie et al., 2019; Girdhar et al., 2020; Lin et al., 2022) have also been used for model-based optimization of tool-aided control. These methods assume that a helpful tool is already present in the scene, whereas we focus on optimizing tool design in conjunction with learning manipulation, which is a more likely scenario for a generalist robot operating for example in a household.

**Joint optimization of morphology and control.** One approach for jointly solving tool design and manipulation problems is formulating and solving nonlinear programs, which have been shown to be especially effective at longer horizon sequential manipulation tasks (Toussaint et al., 2018; 2021). In this work, we aim to apply our framework to arbitrary environments, and so we select a purely learning-based approach at the cost of increasing the complexity of the search space.

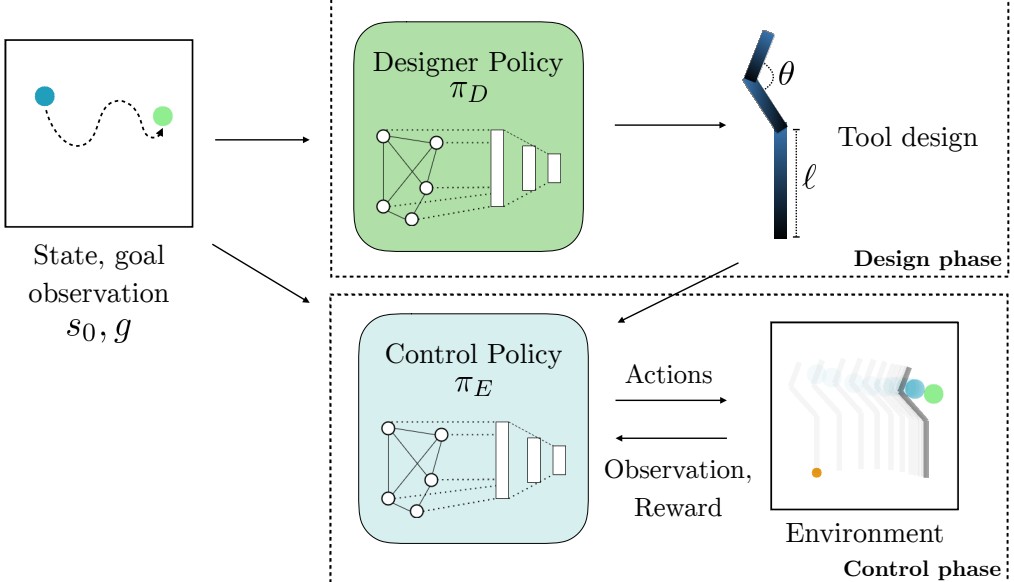

Figure 2: Solving a task using learned designer and controller policies. During the design phase, the designer policy outputs the parameters for a tool that will help solve the given task. In the control phase, the controller policy then outputs motor commands given the tool structure and task specification.

Reinforcement-learning based approaches have also been applied to jointly learning morphology and control. These include policy gradient methods with either separate (Schaff et al., 2019) or weight-sharing (Ha, 2018) design and control policies, or considering the agent design as a computational graph (Chen et al., 2020). Evolutionary or Bayesian optimization algorithms have also been combined with control policies learned through RL to solve a variety of locomotion and manipulation tasks (Bhatia et al., 2021). Luck et al. (2019) use an actor-critic RL formulation with a graph neural network (GNN) value function for improved sample efficiency. Pathak et al. (2019) learn to design modular agents by adding morphology-modifying actions to a MDP. Yuan et al. (2022) provide a generalized formulation with a multi-stage MDP and GNN policy and value networks. These methods have demonstrated promising performance on locomotion tasks. In this work, we focus on manipulation, which often involves sparser rewards and where adding additional actuated joints to designs is infeasible. Furthermore, rather than optimizing for a single task objective, our work addresses the challenge of *goal-conditioned* design and control.

## 3 OUR FRAMEWORK

In this section, we introduce our problem formulation and describe one instantiation of our framework that we study in the remainder of this paper.

### 3.1 PROBLEM SETTING

Our framework tackles learning tool design and use for agents to solve manipulation problems, without any supervision about the quality of designs except for task progress. Because we would like flexibility in the formulation of the design space in an arbitrary control state space, we represent the agent's environment as a two-phase Markov decision process (MDP), including a design phase and an control phase. Then, by jointly learning policies for the two phases, we can use the collected environment interactions for training both policies.

At the start of every episode, the environment begins in the **design phase**, visualized in the top of Figure 2. During the design phase, the action $a \in \mathcal{A}_D$ specifies the parameterization of the tool that will be used for the rest of the episode. In this phase, each state $s \in \mathcal{S}_D$ consists of a vector of task observations, for example positions and velocities of objects in the scene. Because the utility of a given tool varies depending on the task at hand, no rewards are provided during this phase.

After a single transition, the MDP switches to the **control phase**, illustrated in the lower half of Figure 2. During the execution phase, the actions $a \in \mathcal{A}_E$ represent motor commands applied to the previously designed tool, and the agent receives rewards based on task progress (e.g., the distance of an object being manipulated to the target position). The control phase state space $\mathcal{S}_D$ includes the task observations just like in the design phase, but also the tool parameterization from the design phase. The execution phase continues until the task is solved or a time limit is reached, and the episode then ends.

Given this two-phase MDP, the goal is to learn optimal design and control policies, $\pi_D^\star(a|s)$ and $\pi_E^\star(a|s)$ respectively, that maximize the expected discounted return over the entire episode. This is a challenging problem setting because the agent only receives signal about the quality of the tool designs through the final task performance.

In this work, we design this formulation for jointly learning design and control for manipulation. A similar formulation has been explored in prior work for locomotion (Yuan et al., 2022); however, manipulation and locomotion problems have a key difference: while in locomotion, a morphology can be optimized for a single objective like forward velocity and eventually constructed in the real world, tool use for manipulation requires varied design and control strategies based on the task at hand. Therefore, rather than finding a single tool design, we would like to learn a *designer policy* that can create tools depending on the given task, as well as a *controller policy* that can perform manipulation using different tools. We formulate this as conditioning on a supplied goal $g$ from a goal space $\mathcal{G}$, such as the desired final location of an object to be manipulated. The objective is then to find the optimal *goal-conditioned* designer and controller policies $\pi_D^\star(a|s,g)$, $\pi_E^\star(a|s,g)$ that maximize the expected discounted return of a goal-dependent reward function $R(s,a,g)$.

## 3.2 INSTANTIATING OUR FRAMEWORK

Next, we provide a concrete implementation of our framework catered towards solving a series of 2D manipulation tasks. Specifically, we select a **tool design space**, **policy learning procedure**, and **auxiliary reward function**.

**Tool design space.** The design space can significantly impact the difficulty of the joint design and control optimization problem. When the set of possible designs is large but many of them are unhelpful for *any* task, the reward signal for optimization is sparse. Thus, we would like to select a design parameterization that is low-dimensional, but can also enable many manipulation tasks. Furthermore, we prefer designs that are easy to deploy in the real world.

For this implementation, we parameterize the tool using a 3-link rigid chain, but we note that our framework is not limited to this choice of design space. We find that while this parameterization is simple, it is sufficient to help solve a variety of 2D manipulation tasks. Because of its topology, it can also be easily deployed in the real world for example on soft robots (Exarchos et al., 2022) or through rapid fabrication techniques like 3D printing. Specifically, each tool is parameterized by a vector $[l_1, l_2, l_3, \theta_1, \theta_2] \in \mathbb{R}^5$, where $l$ represents each link length and $\theta$ represents the relative angle between the links. The width of each link is fixed.

**Policy learning.** We follow a similar procedure as Yuan et al. (2022) to learn the designer and controller policies. Specifically, we interactively collect experience in the environment using the design and control policies, where each trajectory spans the design and execution phase. We then train the policies jointly using proximal policy optimization (PPO) (Schulman et al., 2017), a popular off-the-shelf policy gradient method. As prior work (Yuan et al., 2022) has shown that graph neural network (GNN) architectures can be helpful for accelerating learning when an input morphology can be converted to a graph representation, we use GNNs as part of our policy architectures for the design, control, and value function networks. The input graph structure consists of a node for each joint in the tool design and an edge for each pair of connected joints. For our chain link parameterization, this is always the three-node path graph $P_3$. Each input node feature includes the environment observation, and additionally for the control policy and value function, the design parameters (e.g. relative angle and length) of the corresponding node in the tool design. As in prior work, we use GraphConv (Morris et al., 2019) to perform graph convolution, before flattening features from all nodes into a single vector and passing this through a multi-layer perceptron (MLP). Additional architectural and training details are provided in Appendix A.

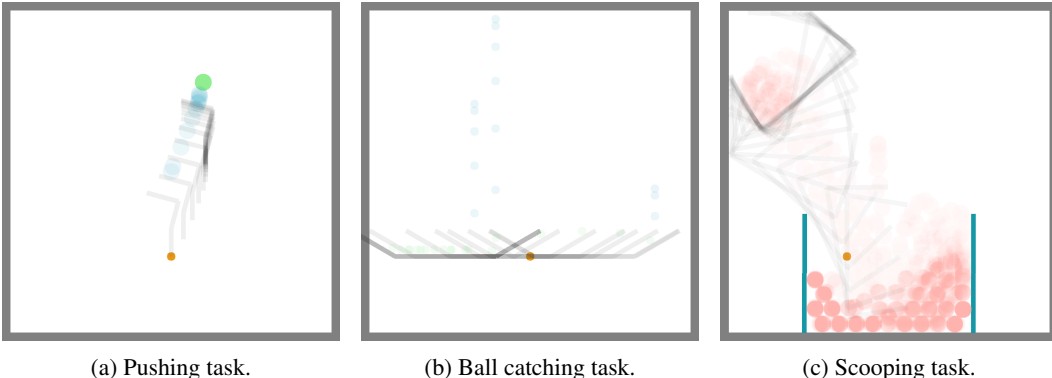

(a) Pushing task.        (b) Ball catching task.        (c) Scooping task.

Figure 3: Visualizations of the 2D simulated manipulation environments for testing our framework.

During training, in order for the designer and controller to learn to achieve a variety of goals, we supply the policies with randomized goals sampled from the environment for each interaction episode. The training objective is to maximize the expected return across sampled goals.

**Auxiliary reward.** When developing embodied agents that can flexibly create and use tools in the real world, resource constraints are an additional consideration. Most co-optimization procedures assume that material such as actuated joints or body links can be arbitrarily added to the agent morphology. However, as an example, when an agent solves manipulation tasks in a household environment, it may not have access to additional motors or large quantities of building materials. One possible solution is for the agent to instead design a tool that requires less material to create, and execute a more complex manipulation trajectory instead. On the other hand, when constraints allow, constructing a larger tool can reduce the amount of energy that the agent consumes through motor control, especially if a task must be completed many times.

We enable our framework to accommodate preferences in this trade-off between design material cost and control energy consumption using a parameter $\alpha$ that adjusts an auxiliary reward that is added to the task reward at each environment step:

$$r_{\text{tradeoff}} = K\left[1 - \left(\frac{\alpha \cdot d_{\text{used}}}{d_{\text{max}}} + \frac{(1-\alpha) \cdot c_{\text{used}}}{c_{\text{max}}}\right)\right]. \tag{1}$$

where $K$ is a scaling hyperparameter that we set to 0.7 for all experiments, $\alpha \in [0, 1]$ controls the balance of emphasis on either the control or design component, $d_{\text{used}}$ and $d_{\text{max}}$ represent the used and maximum possible combined length of the three links in the design respectively, and $c_{\text{used}}$ and $c_{\text{max}}$ represent the control velocity at the current step and the maximum single-step control velocity allowed by the environment. By adjusting the parameter $\alpha$, we can encourage the policy to favor using less material for tool construction when $\alpha$ is large, and less energy for the control policy when $\alpha$ is small.

## 4 EXPERIMENTS

We conduct experiments using three environments created in the Box2D simulator (Catto): pushing, ball catching, and scooping. We select these manipulation tasks to showcase the advantages of using different tools to achieve different goals, when there does not exist a single optimal tool for all goals. The three multi-goal tasks are shown in Figure 3. For each task, we initialize the designed tool by matching a fixed point on the tool to a fixed starting position regardless of the goal. During the control phase, we emulate a scenario in which the tool has been grasped by a robot and is manipulated using velocity control in Cartesian space. A short description of each task is as follows:

- **Pushing**: Push a round puck using the tool such that it stops at the specified goal location. The goal space is a subset of $2D$ final puck locations $\mathcal{G} \subset \mathbb{R}^2$, and the control action space $\mathcal{A} \in \mathbb{R}^2$ specifies the $x$ and $y$ tool velocities. The reward at each step is the change in $\ell_2$ distance between the puck and the goal position after taking that step, plus a bonus of 3.0 for reaching the goal with a velocity below 1.0.

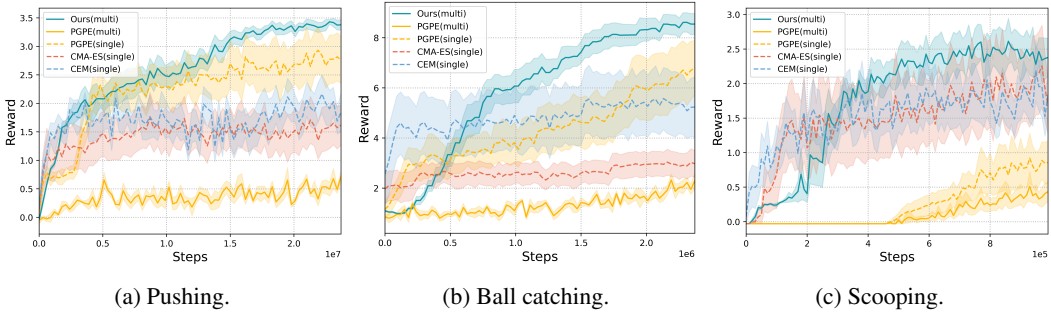

Figure 4: Learning curves for our framework and black-box optimizer baselines on multi-goal and single-goal settings. As baselines, we present the *single-goal* performance of CEM, CMA-ES, and PGPE, as well as the multi-goal training performance of PGPE. We can see that across all studied tasks, our framework achieves substantially improved performance, as well as competitive sample efficiency. Shaded areas indicate standard error across multiple seeds (5 seeds on multi-goal methods and 10 seeds on single-goal methods).

- **Catching**: Use the tool to catch three balls that fall from the sky. The agent's goal is to catch all three balls, which start from varying locations on the $x$-$y$ plane. We use a 1-dimensional control action space that specifies the $x$ velocity of the tool at each step. The reward function is 3.0 for each ball caught, plus a bonus of 3.0 upon capturing all 3 balls.

- **Scooping**: Use the tool to scoop balls out of a reservoir containing 40 total balls. Here we specify goals of either scooping a single ball, or as many as possible. The control action space $\mathcal{A} \in \mathbb{R}^3$ specifies the velocity of the rigid tool in $x$ and $y$ directions, along with its angular velocity. The reward function is a bonus of 6.0 for successfully completing the single ball goal, or a bonus of $\frac{3n}{40}$ for scooping $n$ balls in the multi-ball goal.

In our experiments, we evaluate whether the instantiation of our framework on these $2D$ manipulation tasks has the following three properties that we argue are desirable for embodied agents performing manipulation for tool use:

- Can our framework jointly learn design and control in a sample-efficient manner, using just rewards based on task progress?

- Do our learned designer and controller policies generalize or enable finetuning to solve goals previously unseen during training?

- Can our adjustable parameter $\alpha$ enable agents to specify preferences in the tradeoff between design and control?

## 4.1 EVALUATING SAMPLE EFFICIENCY

First, we investigate whether our framework demonstrates improved sample efficiency for manipulation problems compared to black-box optimization procedures. We compare to the following baseline methods:

- **CEM (single):** We evaluate the cross-entropy method (Rubinstein, 1999; de Boer et al., 2005), a zero-order optimization method that leverages importance sampling to guide the sampling distribution towards higher-scoring values. We evaluate this method in a single-goal setting, optimizing a separate set of parameters for each goal. We formulate the search space as a single vector containing the design parameterization concatenated with control actions for *all* steps in an episode.

- **CMA-ES (single):** We evaluate the covariance matrix adaptation evolution strategy (CMA-ES) (Hansen & Ostermeier, 1996), a popular evolutionary strategy that maintains a population of sampled candidates and models the covariances between each feature of sampled vectors. We also evaluate this method in a single-goal setting, optimizing over the same search space as CEM.

- **PGPE (single):** We evaluate **policy gradient parameter exploration (PGPE)** (Sehnke et al., 2008) in a single goal setting. PGPE is a policy-gradient RL method that uses the

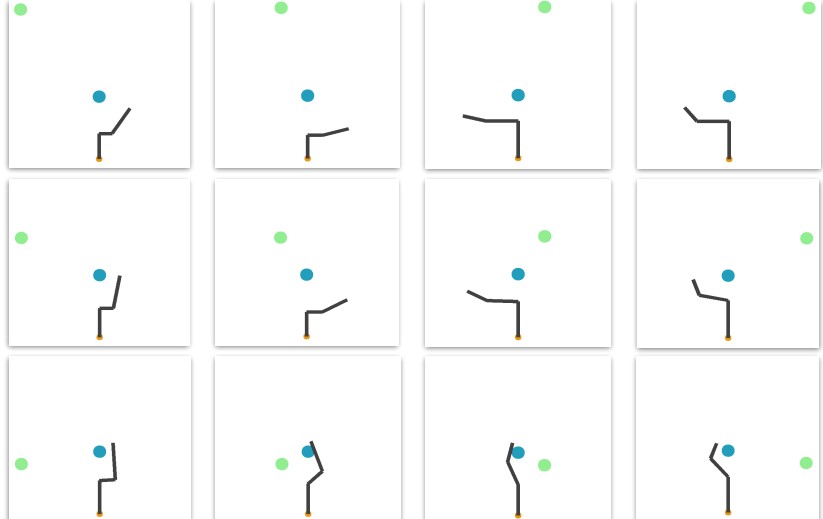

Figure 5: Visualizations of tool designs outputted by a single learned designer policy for the pushing task as the goal position, indicated in green, varies. We can see that the designer outputs a range of tool designs that enable it to more easily push the ball in the desired direction.

natural gradient. We directly optimize the parameters of a neural network that takes as input a single goal observation and the environment state and outputs both the design and control actions. The appropriate action is selected based on the current phase when rolling out the policy in the environment.

- **PGPE (multi):** We additionally compare against PGPE in a multi-goal setting with a similar setup as in the single goal setting but instead train a goal-conditioned neural network to output design and control parameters.

We implement each baseline method using the EvoTorch library (Toklu et al., 2022). Additional hyperparameters and tuning details for our comparisons can be found in Appendix A.1.

In Figure 4, we compare the learning curves of our method and the baselines for all three tasks. The learning curves show the total reward versus the number of environment steps taken for each method. Compared to the single-goal baselines, we see that training using our framework exhibits better final performance and competitive sample efficiency. However, we emphasize that the single-goal baselines optimize one model for each goal, while our method learns one model for *all* goals. When compared to PGPE (multi), which learns one model for multiple goals, our method outperforms this baseline in both sample efficiency and final performance.

We present qualitative examples of tool designs output for different goals on the pushing task in Figure 5. We find that our designer policy outputs a diverse range of tools depending on the specified goal location, and the controller policy is able to perform manipulation conditioned on the tool design and goal to solve the task.

## 4.2 GENERALIZATION TO UNSEEN GOALS

Rather than learning a single design and a policy to control it, our framework seeks to enable rapid tool prototyping for manipulation by learning goal-conditioned designer and controller policies. In simulation, these policies can experience millions of trials for a range of goals. However, when deployed to the real world, designer and controller policies cannot be pre-trained on all possible future manipulation goals. The agent may instead leverage what it has learned about tools that can be useful for an unseen task from the goals it has seen. In this section, we test the ability of our policies to generalize to goals unseen during training. Because it has a goal space that can be manipulated in a semantically meaningful way, we focus on the pushing task for these experiments. We train policies using our framework on a subset of goals from the entire goal space by removing a region of the space, which we call the "cutout" region. (see Figure 6a). For the pushing task, because the goal space is the rectangular region defined by $x \in [8.0, 32.0]$ and $y \in [18.0, 34.0]$ specifying

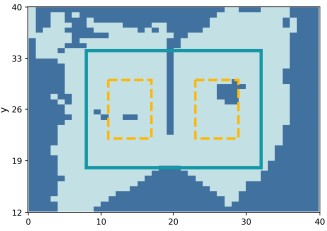
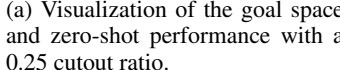
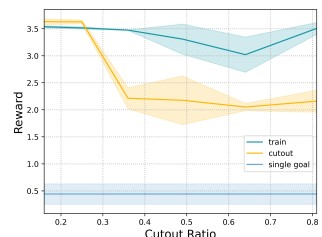
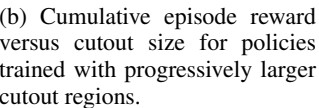
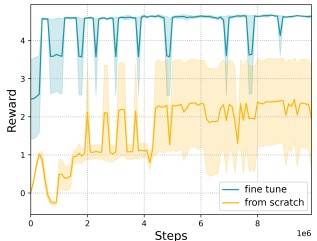

(a) Visualization of the goal space and zero-shot performance with a 0.25 cutout ratio.

(b) Cumulative episode reward versus cutout size for policies trained with progressively larger cutout regions.

(c) Fine-tuning performance compared to learning from scratch across 3 target goals.

Figure 6: Interpolation results on the pushing task. In (a), we plot the success (light blue) and failure (dark blue) goal regions. Areas within the dotted yellow borders denote unseen cutout regions (interpolation). The region within the teal border (but outside cutout regions) shows the training region. The area outside the teal border is unseen during training (extrapolation). (b) and (c) show reward curves (averaged over 3 runs). Shaded regions denote standard error. In (b), we observe that the performance for policies trained with small cutouts is close to that of the setting trained on all goals. In (c) we show evaluation performance of policies finetuned for a specific unseen goal compared to learning curves for single-task policies with those goals.

desired final ball location, the cutout region corresponds to two disconnected rectangular patches in the goal space with $x_1 \in [11.0, 17.0]$, $y_1 \in [22.0, 30.0]$ and $x_2 \in [23.0, 29.0]$, $y_2 \in [22.0, 30.0]$ respectively. Then, we evaluate the generalization performance of learned policies on the goals from the cutout region and outside the training region in two separate scenarios.

**Zero-shot performance.** In the first scenario, we test the ability of the designer and controller policy to tackle a previously unseen goal directly. Using a policy trained on the goal space with a cutout region removed as described above, we evaluate the zero-shot performance by randomly sampling unseen goals. In Figure 6a, we visualize the zero-shot performance of our design and control policies on goals across the entire environment plane, finding that our policies are able to solve even goals outside the training region boundaries.

Next, we analyze how decreasing the number of possible training goals affects generalization performance. We train six policies using our framework where the cutout region removes a fraction of the total training area equal to $0.16, 0.25, 0.36, 0.49, 0.64$, and $0.81$ respectively. We then plot the performance of these learned policies on goals seen and unseen during training.

In Figure 6b, we show the returns as the size of the cutout region for training goals varies. As a reference, we additionally visualize the performance of policies trained with our framework to solve a single randomly selected task. When the cutout region is very small, the performance of our learned policies on seen and unseen goals is similar. As the area of the cutout goal region increases, the performance on unseen goals drops, but is still much better than the single goal baseline and is able to solve a significant portion of unseen tasks.

**Finetuning performance.** Sometimes, new goals cannot be solved by directly applying our pre-trained designer and controller policies. In this section, we test whether our policies can still serve as good instantiations for achieving these goals. We hypothesize that even when our policies do not solve the task directly, the designer policy may still be able to propose a reasonable tool that is helpful for solving the task, while the controller policy will be able to generalize to manipulate new tools. We test this by starting with policies pre-trained with our framework on the entire training goal region and finetuning them to solve goals outside that region. Specifically, we select three goals for finetuning: $\{(2, 13), (37, 28), (37, 13)\}$. For comparison, we also train three separate policies using our framework to solve the *single-goal* pushing task towards each of these goals.

In Figure 6c, we show the results of the finetuning experiment. We find that even for goals that are far away from the initial training region, our policies are able to learn to solve the task within a handful of gradient steps. Because the pre-trained policies are able to find reasonable control and especially tool initializations, the joint space of designs and control policies to explore is significantly narrowed. In comparison, learning a single-goal policy from scratch is challenging because achieving these distant goals requires many consecutive near-optimal decisions.

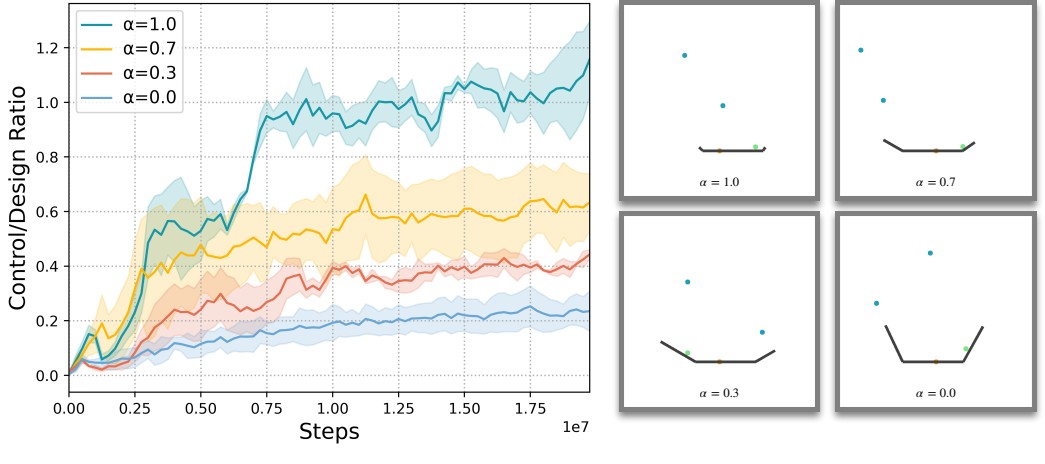

(a) Control/Design ratio with different $\alpha$.

(b) Tools produced by designer policies at different levels of $\alpha$.

Figure 7: Qualitative examples of tools generated by setting our tradeoff parameter $\alpha$ to different values. We can see that as $\alpha$ increases, the tools created by the designer policy have shorter links at the left and right sides to decrease material usage. With low $\alpha$ values, large tools prevent the control policy from having to move the tool far.

### 4.3 TRADING OFF DESIGN AND CONTROL COMPLEXITY

In this section, we aim to determine whether our introduced tradeoff parameter $\alpha$ is effective at actualizing preferences in the tradeoff between design material cost and control energy consumption. For this experiment, we focus on the **catching** task, because the tradeoff has an intuitive interpretation in this setting: a larger tool can allow the agent to catch the objects with minimal movement, while a smaller tool can save on material cost but requires a longer trajectory that has additional energy costs. We train four agents independently on the catching task, setting the value of $\alpha$, the tradeoff reward parameter defined in Equation 1, to 0, 0.3, 0.7, and 1.0 respectively.

In Figure 7a, we track the relative amounts of energy expended for control and material used for design across these agents. Specifically, we plot the ratio $\frac{d_{\text{used}}/d_{\text{max}}}{c_{\text{used}}/c_{\text{max}}}$, where $d$ represents the combined length of all tool links and $c$ is the per-step control velocity. $d_{\text{max}}$ and $c_{\text{max}}$ indicate the maximum tool size and control velocity allowed by the environment. We find that this ratio indeed correlates with $\alpha$, which indicates that agents that are directed to prefer saving either material or energy are doing so, at the cost of the other. We also visualize the outputted tool designs in Figure 7b. We see that for progressively larger values of $\alpha$, the agent shortens the tool length on the side of the catcher to conserve material, and instead navigates the tools further across the plane to catch the balls.

## 5 CONCLUSION

We have introduced a framework for agents to jointly learn design and control policies with the purpose of solving manipulation tasks. Because the best type of tool and control strategy can vary widely depending on the manipulation goal, we propose to learn designer and controller policies to generate useful tools based on the task at hand and perform manipulation with them. By leveraging reinforcement learning methods that use GNNs to compute features, we demonstrate using simulated 2D manipulation tasks that our framework can be instantiated to learn designer and controller policies for a variety of commanded goals. We find that when training on only a subset of possible goals, the learned policies can perform zero-shot generalization or rapid finetuning using many fewer samples compared to learning from scratch. We finally demonstrate that, through the addition of a tunable tradeoff parameter, the learning procedure can be adjusted to better suit practical requirements such as resource constraints.

**Reproducibility.** We described our method and experimental setup in detail in Sections 3 and 4. The code will be released upon publication to facilitate future research.

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

# A  TRAINING HYPERPARAMETERS & ARCHITECTURE DETAILS

## A.1  OUR FRAMEWORK

Here we provide detailed hyperparameters for our framework. Unless otherwise specified, we use the neural network architectures for the design policy, control policy, and value function from Yuan et al. (2022).

| Hyperparameters | Values |
|---|---|
| Tool Position Init. | (20, 10) |
| Control Steps Per Action | 1 |
| Max Episode Steps | 150 |
| Slack Reward | -0.001 |
| Tool Length Ratio | (-0.5, 0.5) |
| Tool Length Init. | (2.0, 2.0, 2.0) |
| Tool Angle Init. | (0.0, 0.0, 0.0) |
| Tool Angle Ratio | (-1.0, 1.0) |
| Tool Angle Scale | 90.0 |
| Control Log Std. | -1.0 |
| Design Log Std. | -2.3 |
| Fix Design & Control Std. | True |
| Policy Learning Rate | 2e-5 |
| Entropy $\beta$ | 0.01 |
| Value Learning Rate | 1e-4 |
| KL Divergence Threshold | 0.005 |
| Batch Size | 50000 |
| Minibatch Size | 2000 |
| PPO Steps Per Batch | 10 |

Table 1: Hyperparameters used for our framework on the pushing task.

| Hyperparameters | Values |
|---|---|
| Tool Position Init. | (20, 10) |
| Control Steps Per Action | 1 |
| Max Episode Steps | 150 |
| Slack Reward | -0.001 |
| Tool Length Ratio | (-0.5, 2.0) |
| Tool Length Init. | (2.0, 1.0, 1.0) |
| Tool Angle Init. | (0.0, 0.0, 0.0) |
| Tool Angle Ratio | (-1.0, 1.0) |
| Tool Angle Scale | 60.0 |
| Control Log Std. | 0.0 |
| Design Log Std. | 0.0 |
| Fix Design & Control Std. | True |
| Policy Learning Rate | 2e-5 |
| Entropy $\beta$ | 0.01 |
| Value Learning Rate | 1e-4 |
| KL Divergence Threshold | 0.002 |
| Batch Size | 50000 |
| Minibatch Size | 2000 |
| PPO Steps Per Batch | 10 |

Table 2: Hyperparameters used for our framework on the ball catching task.

## A.2  BASELINES

For each baseline, we perform hyperparameter sweeps for a fair comparison with our framework. The tested hyperparameter configurations for each baseline are listed in Table 4.

| Hyperparameters | Values |
|---|---|
| Tool Position Init. | (15, 10) |
| Control Steps Per Action | 5 |
| Max Episode Steps | 30 |
| Slack Reward | -0.001 |
| Tool Length Ratio | (-0.7, 0.2) |
| Tool Length Init. | (6.0, 3.0, 3.0) |
| Tool Angle Init. | (0.0, 0.0, 0.0) |
| Tool Angle Ratio | (-0.1, 0.7) |
| Tool Angle Scale | 90.0 |
| Control Log Std. | 0.0 |
| Design Log Std. | 0.0 |
| Fix Design & Control Std. | True |
| Policy Learning Rate | 2e-5 |
| Entropy $\beta$ | 0.01 |
| Value Learning Rate | 3e-4 |
| KL Divergence Threshold | 0.1 |
| Batch Size | 10000 |
| Minibatch Size | 400 |
| PPO Steps Per Batch | 10 |

Table 3: Hyperparameters used for our framework on the scooping task.

| Method | Hyperparameters | Values |
|---|---|---|
| CMA-ES | Population Size | 10, **24**, 100, 1000 |
| | Initial Stdev | **0.1**, 1.0, 10.0 |
| | Center Learning Rate | 0.01, 0.1, **1.0** |
| | Covariance Learning Rate | 0.01, 0.1, **1.0** |
| | Rank $\mu$ Learning Rate | 0.01, 0.1, **1.0** |
| | Rank One Learning Rate | 0.01, 0.1, **1.0** |
| CEM | Population Size | 10, **24**, 100, 1000 |
| | Initial Stdev | **0.1**, 1.0, 10.0 |
| | Parenthood Ratio | 0.01, **0.1**, 0.4, 0.8 |
| PGPE | Population Size | 10, **24**, 100, 300 |
| | Center Learning Rate | 0.001, **0.01**, 0.1, 1.0 |
| | Stdev Learning Rate | 0.001, **0.01**, 0.1, 1.0 |
| | Initial Radius | 0.01, 0.1, **1.0**, 2.0, 5.0 |

Table 4: We tune over these values for hyperparameters of baseline methods. Bolded values indicate the best performing settings, which we use in our comparisons.

