# OpenReview forum: "Designing and Using Goal-Conditioned Tools"
_ICLR.cc/2023/Conference — Submitted to ICLR 2023_

### Official Review · Reviewer_kbaD · 2022-10-23

**Confidence:** 4
**Correctness:** 3
**Technical Novelty And Significance:** 2
**Empirical Novelty And Significance:** 2
**Recommendation:** 5

**Clarity, Quality, Novelty And Reproducibility:**

The paper is clearly written, and the approach seems simple enough to reproduce. I think the proposed problem is somewhat novel, but the solution is relatively basic and doesn't necessarily advance the field substantially because of the constrained nature of the tasks being solved.

**Strength And Weaknesses:**

Strengths:

The paper clearly motivates the need to equip agents with ability to design and use tools (constrained by available materials), and I agree this is an interesting manipulation task. The argument that unlike locomotion, where it suffices to learn and optimise the policy of a single morphology, manipulation with tools requires the ability to design tools is a good one, and I think this will inspire an interesting line of work.

Weaknesses:

Unfortunately, tool construction is neglected by this work, which rapidly simplifies tool design to a problem of regressing the fixed parameters of a highly constrained model (3 link) and then learning a general policy to solve talks requiring relatively simple motion trajectories (using much the same approach as in the locomotion setting, which contradicts the argument that this is somehow different for the tool use case).

The work only considers very toy problems (Box2d envs), and glosses over the challenge of actually assembling a tool prior to use. I understand that abstractions like this are important to progress algorithmically, but this abstraction and policy choice very conveniently fits existing approaches [Yuan 22] and the fixed 3-link tool removes some of the really interesting aspects of what it means to design a tool with interesting affordances from available materials (eg. the macgyvering approach of Nair et al.). It is also unclear to what extent the proposed approach will scale.

Ultimately, this means the experimental results amounts to a comparison between the design, control approach and evolutionary and policy gradient parameter estimation techniques in a very constrained environment, which while interesting, does not really line up with the ultimate motivation of this work.


**Summary Of The Paper:**

This paper introduces a new task of joint tool and controller design. Here policies are required to first produce a tool, and then to use this to solve some goal oriented task. In this work tool production policies essentially predict the parameters of a 3-link chain (angles and lengths), which is then used by the control policy. Results show that this two stage policy optimisation process is more effective than a range of evolutionary and parameter search strategies. The idea is interesting and the motivation very exciting, but the final results slightly underwhelming.

**Summary Of The Review:**

Overall, I like the problem set out in the motivation, and think this is interesting. Unfortunately, the solution proposed in this work seems too constrained to have much practical value, and it is unclear that the proposed approach and findings will scale to more complex tool design settings.

---

> ### Author Response · Authors · 2022-11-18
> **Response to Reviewer kbaD**
>
> Thank you for your detailed review and constructive feedback. We are delighted to learn that you liked and agreed with our motivation to equip agents with the ability to design and use tools, how our tool manipulation setup differs from locomotion tasks in prior works, and that you think our contributions will inspire an interesting line of work. We address your questions below.
>
>
> > Tool construction is neglected by this work, which simplifies tool design and learns a general policy to solve tasks using much the same approach to the locomotion setting.
>
> We agree that tool construction is an important part of a real-world tool design and use pipeline, and in this work, we do not explicitly focus on that aspect.
>
> We choose to compose our tool design with primitive shapes with the ease of manufacture/construction in mind. We also note that our framework is not constrained to this particular choice of tool design parameterization, and rather only uses it as a proof of concept. The focus of the paper is the joint tool design and control problem and framework it poses, rather than a state-of-the-art method that already solves the problem for tasks requiring complicated tool designs and controls.
>
> In the locomotion tasks considered in prior work, the final control policies are conditioned on only a single design optimized for a single task, whereas our tool manipulation tasks require design-conditioned control policies.
>
>
> > The work considers toy problems and the fixed 3-link tool is limited; it is not clear that the proposed approach and findings will scale to more complex tool design settings.
>
> We choose the 3-link tool design space to reduce the unnecessary design space that corresponds to tool designs that are unhelpful for any task, while still enabling a wide variety of manipulation tasks. Our framework is not limited to the particular design space of 3-link tools, but rather uses the tools and tasks in this paper as a proof of concept to demonstrate that our framework can solve 2D manipulation tasks by jointly optimizing tool design and control.
>
> We appreciate your constructive feedback and aim to demonstrate the method on more diverse tool design parameterizations and more complex, for instance, 3D tasks in the future.

---

> > ### Comment · Reviewer_kbaD · 2022-11-23
> > **Thanks for your response**
> >
> > Thank you for your rebuttal.

---

### Official Review · Reviewer_xbGt · 2022-10-24

**Confidence:** 1
**Correctness:** 3
**Technical Novelty And Significance:** 3
**Empirical Novelty And Significance:** 3
**Recommendation:** 5

**Clarity, Quality, Novelty And Reproducibility:**

	- I think the paper proposes novel contributions, but my understanding may be very limited.

I do have concerns on the reproducibility of this work though, since it is not clear how these tools can be used off the shelf by existing practioners. It might be useful if the paper was written in such a way to show how the tools can be used - although I am not sure whether this contribution is ideal for a ICLR submission or not.

**Strength And Weaknesses:**

1. The paper is a bit difficult to understand in its current form, and it is not clear how these policies are actually designed. It may however be due to my lack of understanding or familiarity with contributions of this type.
2. The experimental demonstrations considered are well explained - but the exact framework in the way this paper is written, is difficult to understand.
3. The paper seems to propose a useful contribution, especially based on the claim that the proposed framework and tool can be used for tackling out of distribution goals too. The overall pipeline is a bit difficult to understand, and it would have been useful if a schematic of the overall pipeline could be provided.
4. I do think this type of work is useful for the community, especially for steps towards making RL useful in the real world.

**Summary Of The Paper:**

This paper provides a tool to design policies for solving tasks for different goals. It provides a general framework, that the downstream policy can then use in a RL framework. The paper provides proof of concept demonstrations for designing policies able to solve different manipulation tasks, conditional on the type of goal being used.

**Summary Of The Review:**

My understanding of this paper is very limited and I am not sure if my review should be counted as fair. It is very much possible that the core contribution of this work is beyond the scope of my understanding. The work seems useful, especially if the tools can be used by RL practitioners off the shelf.

---

> ### Author Response · Authors · 2022-11-18
> **Response to Reviewer xbGt**
>
> Thank you for your thoughtful review and feedback. We are glad to hear that you find our contributions useful to help the community move RL towards real-world applications and that our experiments are well explained. We respond to your questions below.
>
> > Specifics of how the policies are designed and the overall pipeline schematic
>
> Thank you for your feedback on the presentation of the overall pipeline to help improve the clarity of the technical sections of our paper. We would much appreciate any additional specific feedback and suggestions to make the pipeline more clear!
>
> Our proposed framework tackles the problem of jointly learning a goal-conditioned designer policy and a design-agnostic controller policy. We provide in the paper an illustration of our proposed framework in Figure 2.
>
> Our framework consists of two phases: a design phase, and a control phase.
>
> In the design phase, a learned designer policy generates tool design parameters that will help solve the given task. In the control phase, a learned controller policy outputs motor commands that control the designed tool to solve the task. We optimize the designer and the controller policies jointly with proximal policy optimization (PPO).
>
> > Reproducibility and how the tools can be used
>
> To ensure the reproducibility of our results, we described our method and experimental setup in detail in Sections 3 and 4. The code will be released upon publication to facilitate future research.
>
> Regarding how the tools can be used, in this work, we focus on a 3-link tool design parameterization as a first step towards demonstrating the ability of our proposed framework of solving manipulation tasks. We aim to demonstrate the method on more diverse tool design parameterizations and more complex, 3D tasks in the future.

---

### Official Review · Reviewer_BpS2 · 2022-10-25

**Confidence:** 4
**Correctness:** 3
**Technical Novelty And Significance:** 2
**Empirical Novelty And Significance:** 2
**Recommendation:** 3

**Clarity, Quality, Novelty And Reproducibility:**

The work is presented clearly - but the introduction could do with some clarification following the comments above. As mentioned above, the novelty is in extending existing work to the goal-conditioned setting. However, there is a lack of focus and explanations in the methods regarding this.

**Strength And Weaknesses:**

Strengths:
- The novelty of this work lies in extending a previous framework of morphology design and control to the goal-conditioned setting.
- Proven building blocks which handle this problem well such as GNNs are used.
- The auxiliary reward used for a tradeoff between material cost and control seems like something practical.


Weaknesses:
- In the introduction, it is not clear what a task/goal is and sometimes this is used interchangeably which can be confusing. The authors work here corresponds to a single task where multiple goals are possible. However, when reading the introduction, I had the impression that the authors were proposing a framework to deal with multiple tasks with one policy. For example, “design appropriate tools when presented with a task”.  But the experiments clearly show it is for dealing with multiple goals (i.e. number of balls) for the same task (i.e. scooping).
- In section 3.2, it is slightly weird to say the proposed framework is “not limited to the selected design space choice”, given that the authors themselves explain why a low-dimensional parameterisation is used during the design stage to avoid sparse rewards during optimisation.
- The novelty of this work lies in extending a previous framework to the goal-conditioned setting. However, I find not many details on how this affected the need to change anything in the methods from prior work to accommodate for this. It seems as if the goals were just additionally supplied to the policy with randomised goals as in normal GC-RL. Would appreciate a bit more explanation if any additional things were required, and if not, why not? Would also appreciate some ablations on what was critical to this method working.
- I find it odd that the pushing task is used in the experiments where the goal space is final puck locations, given that one of the authors' main arguments when motivating the paper is contrasting with locomotion tasks from prior work which usually only involve a single goal. But a locomotion task can also involve moving the various 2-D locations and not just going forward.
- Could the authors provide some explanation on why only black-box optimisation baselines are used? Given that these methods are generally sampling-based methods, they are usually slightly more sample inefficient so comparing a PPO (policy gradient based - authors method) approach to such baselines would not be completely fair. If the authors think this is fair or there is a limitation on why other baselines can’t be used like goal-conditioned PPO, I would appreciate seeing them explained in the paper. Why couldn’t the previous work which the authors build on, be used as a single goal baseline (since the authors are also comparing to other single baselines)? Likewise, why aren’t more “multi” baselines presented?

**Summary Of The Paper:**

Goal: creating specialised tools for different manipulation goals.

Proposed Solution: A designer policy that is conditioned on the task goal + design agnostic controller policy that can perform manipulation using the designed tools. This paradigm has already been proposed in recent work (Yuan et al.) the authors have cited. The authors extend this to the goal-conditioned setting to accommodate multiple goals within a single manipulation task. (i.e. the designer can immediately design task-appropriate tools, control policy can directly use the designed tool for the task)

Designer Policy:
State space consists of observations (positions of velocities of objects in the scene)
Action space consists of the parameterisation of the tool (designed tool)
Rewards: NO rewards provided
One transition is designing one tool in which the MDP switches to the control phase.
In this paper, the tool is parameterised by a 5-dimensional vector representing a 3-link rigid chain (length of each link and the relative angles between links)

Control Policy:
State space consists of the observations + the parameterisation of the tool from the design phase.
Action space consists of the motor commands applied to the designed manipulation tool.
Reward: receives task reward

**Summary Of The Review:**

Extending existing previous work to the goal-conditioned setting seems like a good idea. While the methods seem sound, the experiments and baselines selected and presented and unconvincing and slightly weak for the above reasons listed.

---

> ### Author Response · Authors · 2022-11-18
> **Response to Reviewer BpS2**
>
> Thank you for your thorough review and detailed feedback. We are delighted to hear that you liked the novel problem we posed to jointly learn goal-conditioned tool design and control for manipulation and our idea to extend prior work to the goal-conditioned setting, the practicality in the auxiliary reward we proposed to tradeoff between material and control costs, and that our work is presented clearly. We address your questions below.
>
> > Consistent use of the terminology for a task/goal
>
> Thanks for pointing this out. We use the term “task goal” to denote different variations (for example, target position) of a task, and have updated the paper to consistently use “task goal” instead of “task”. Thank you for your helpful feedback on our manuscript!
>
> > It is slightly weird to say the framework is not limited to the selected design space given that authors explain the particular choice of a low-dimensional parameterization
>
> We choose this particular low-dimensional parameterization to reduce the unnecessary design space that corresponds to tool designs that are unhelpful for any task, while still enabling a wide variety of manipulation tasks. Our proposed framework is generic and not constrained to the particular design space choice. Benefiting from the multi-stage design MDP formulation in Transform2Act, our framework in principle works with an arbitrary number of tool links, primitive shapes, and joint connections.
>
> > Details on how the goal-conditioned setting required changes in the method compared to prior work
>
> While our formulation is implemented in a similar way as standard goal-conditioned RL, we found that training in the multi-goal setting is much less stable compared to a single goal, often causing policy performance to suddenly crash. This is similar to observations from prior work that find challenges in training goal-conditioned RL policies [1]. We address this by implementing early stopping during PPO training based on a KL-divergence target between the policy with updated parameters and the policy at the start of each training epoch.
>
> > The use of the pushing task with the goal space as final puck locations, and its contrast with single-goal locomotion tasks from prior work, which can also involve moving to various 2D locations.
>
> For locomotion tasks with various 2-D locations as goals, the designed morphologies required to achieve those locations can be very similar. In contrast, for our pushing task, the agent needs to produce different designs based on the target by leveraging tool geometry to control the direction of pushing. The pushing task involves more complicated dynamics than “carrying” the puck to a goal location. For instance, the agent can push with a larger force to let the puck move a longer distance than the tool, and can even leverage the environment to bounce the puck off obstacles.
>
> > Why are only black-box optimisation baselines used? These methods are usually slightly more sample inefficient so the comparison to PPO is not completely fair.
>
> Policy gradient parameter exploration (PGPE), one of our baselines, is a policy gradient method, like our PPO-based approach.
>
> > If the authors think this is fair or there is a limitation on why other baselines can’t be used like goal-conditioned PPO, I would appreciate seeing them explained in the paper. …  Likewise, why aren’t more “multi” baselines presented?
>
> The approach proposed in this work is based on PPO with goal conditioning. To our knowledge, our work is among the first to tackle the problem of goal-conditioned joint tool design and control optimization. We presented PGPE (multi) as a baseline for a reasonable off-the-shelf approach, and would be happy to improve the thoroughness of our evaluation with other methods that address the same problem setting.
>
> > Why couldn’t the previous work which the authors build on, be used as a single goal baseline (since the authors are also comparing to other single baselines)?
>
> In this work, we extend the prior work from the single goal to a multi-goal setting to eliminate the need to train a separate agent for each different task goal. We compute the single-goal baseline performances by taking the average of multiple models each trained on one task goal. We included the single goal baselines as proxies for a performance upper bound for these methods rather than as completely fair comparisons.
>
> References
>
>  [1] “Gradient Surgery for Multi-Task Learning.” Tianhe Yu, Saurabh Kumar, Abhishek Levine, Karol Hausman, Chelsea Finn.

---

### Decision · Program_Chairs · 2023-01-20

**Decision:**

Reject

**Justification For Why Not Higher Score:**

None of the reviewers recommends acceptance due to the weaknesses outline above.

**Justification For Why Not Lower Score:**

N/A

**Metareview: Summary, Strengths And Weaknesses:**

(a) Summary
The paper extends a method by Yuan et al. for combining morphology and controller design with design-conditioned control policies. The paper focuses on 2D manipulation tasks with 3 link rigid chains as tools. The proposed approach is compared to a number of baselines.

(b) Strengths
- The paper tackles a very interesting problem
- The paper is well written and easy to follow
- The experimental results clearly show the benefits of the method

(c) Weaknesses
- There are still doubts about the significance of the proposed extension
- As the authors say, the experiments are currently more a proof of concept than showing the full potential of the method (or conversely, it is unclear how well this will scale)

**Summary Of Ac-Reviewer Meeting:**

N/A